# Unlocking the Potential of Virtual Reality in Human Robot Interaction: Insights for User Studies

### Eleonora Zodo
AIT Austrian Institute of Technology GmbH
Vienna, Austria
eleonora.zodo@ait.ac.at

### Katja Gallhuber
AIT Austrian Institute of Technology GmbH
Vienna, Austria
katja.gallhuber@ait.ac.at

### Setareh Zafari
AIT Austrian Institute of Technology GmbH
Vienna, Austria
setareh.zafari@ait.ac.at

### Jaison Puthenkalam
AIT Austrian Institute of Technology GmbH
Vienna, Austria
jaison.puthenkalam@ait.ac.at

### Andreas Sackl
AIT Austrian Institute of Technology GmbH
Vienna, Austria
andreas.sackl@ait.ac.at

### Manfred Tscheligi
AIT Austrian Institute of Technology GmbH
Vienna, Austria
manfred.tscheligi@ait.ac.at

## ABSTRACT

This paper describes the design and implementation of a Virtual Reality (VR) HRI user study, which was used to derive insights and recommendations for researchers who plan to conduct similar VR user studies. Although VR user studies can be an efficient way to evaluate HRI variations before real world implementations are executed, several aspects need to be considered. According to our recommendations, the intentions of the robot need to be transparent for the VR study participants, a controlled test environment is crucial, ergonomic aspects need to be considered and a realistic representation of the work context should be provided. Our recommendations are based on the feedback of skilled workers in the field of ceramic polishing and users with technical background and varying experience with robots.

## CCS CONCEPTS

• **Computer systems organization** → **Robotics**; • **Human-centered computing** → **Virtual reality**; **User studies**.

## KEYWORDS

HRI, subjective experiment, HMLV

ACM Reference Format:
Eleonora Zodo, Katja Gallhuber, Setareh Zafari, Jaison Puthenkalam, Andreas Sackl, and Manfred Tscheligi. 2023. Unlocking the Potential of Virtual Reality in Human Robot Interaction: Insights for User Studies. In *VAM-HRI '23: International Workshop on Virtual, Augmented, and Mixed-Reality for Human-Robot Interactions, Macrh 13, 2023, Stockholm, Sweden*. ACM, New York, NY, USA, 6 pages. https://doi.org/XXXXXXX.XXXXXXX

*VAM-HRI '23, March 13, 2023, Stockholm, Sweden*
© 2023 Association for Computing Machinery.
ACM ISBN 978-1-4503-XXXX-X/18/06...$15.00
https://doi.org/XXXXXXX.XXXXXXX

## 1 INTRODUCTION

In recent years, there has been an increasing interest in the empowering effect of human robot collaboration in industrial contexts. However, while the quality of collaboration and the resulting work performance depend on how well the task interdependency between human and robot is coordinated [20], selecting the best user interaction modalities with the system (e.g. screen interaction vs. gesture-based interactions) is not necessarily straightforward.

In addition, there is a generally low market share of collaborative robots installed in practical industrial settings [7]. A more comprehensive inclusion of end-users in the design phase of such systems could improve this situation, e.g. via user studies. In particular, developing user interfaces for human robot interaction depends not solely on technical challenges but also on human factors and how users experience the interaction [1]. Although insights from user studies provide valuable feedback for implementation, the execution of such studies in the context of human-robot-interaction often requires costly testing stations to interact with. To overcome this limitation, an innovative and cost-efficient approach is the usage of Virtual Reality (VR) setups, which allows end-user to try out interaction modalities without the need to create real-world test installations. We created a VR application which allows end-users to experience different Human-Robot-Interaction (HRI) modalities for collaboration with a polishing robot, in order to determine which modality shall be implemented in an industrial robotic system. In this paper, we focus on user feedback about our VR setup and provide recommendations for future studies of this kind.

## 2 RELATED WORK

In various industrial contexts (e.g., polishing and surface finishing of complex work pieces), it is desired to - at least partially - automate certain repeating and/or time-consuming tasks. Instead of aiming for full automation and delegating all tasks to robots, the focus should be on collaborative approaches where humans and robots work closely together on the same task doing what each of them does best at any specific moment [13]. Thus, collaborative scenarios are required, which need to consider both human and robot capabilities and requirements. During the development phase of such complex systems, design decisions need to be taken,

e.g., how a specific human-robot-interaction is implemented. A well-established way to evaluate design alternatives is to compare different prototypes via user studies in a laboratory or field setting. However, such studies consume a lot of resources, which make alternative evaluations methods such as VR-studies more attractive [2].

In the context of HRI, VR applications are already used in several ways, e.g., for programming, simulation and tele-operation of industrial robots (e.g., [4, 5, 11, 15]). For example, [8] compares proactive and adaptive techniques for safe collaboration between humans and robots in virtual environments. It was shown, that in case of collision with the human, collaboration efficiency is better when the robot slows down its speed (proactive technique) than when it retracts and follows a different path (adaptive technique). However, most of these studies have been restricted to comparisons of virtual environments (such as interface, task, robot) with their physical counterparts (e.g., [6, 9, 16, 18]). For instance, [17] found that interfaces created in VR allowed non-expert users to tele-operate a robot to complete manipulation tasks faster and with lower cognitive workload than the related physical keyboard and monitor interface.

There are also some papers about HRI VR studies which evaluate interaction modalities. For example, [3] evaluated interactions of the user with a robot via two different control modes (prosthetic mimicking and kinematics control) in VR and found that the kinematics condition was easier to use than prosthesis mimicking. In [12], the authors compared two ways of projecting visual cues about the robot's intentions via a mixed reality approach, namely visualizing the robot's upcoming movements vs. highlighting the object that the human needs to interact with, and hypothesized that a combination of these two approaches may lead to better results.

What is missing so far are recommendations for executing VR HRI user studies.

## 3 VR STUDY SETUP

Our VR study investigates and compares different interaction modalities for Human-Robot Collaboration to perform the quality inspection and end-polishing of ceramic basins. Users had to encircle detected defects (like scratches or cracks) on the basins with a pen, to mark them for polishing by the robot. Once all defects had been marked, they would trigger a scanning process that utilises a camera system to detect areas with annotations. The robot would afterwards provide feedback about the areas with markings that have been detected, by projecting a blue rectangle over them and showing buttons to delete or modify these polishing areas. This way, users were provided with a way to correct possible mistakes and to adjust the parameters of the polishing process (e.g. pressure, trajectory, etc.). After a final confirmation by the user, the polishing process was executed by the robot.

We conducted a within-subjects study to compare two interaction paradigms: interactions directly on the work piece vs. a separate monitor. Each participant evaluated four conditions: C1) users both mark the defects and edit the detected areas on the work piece surface, C2) users mark the defects on the work piece surface, but the detected areas are displayed on a monitor where users can edit them, C3) users mark the defects on a monitor, the detected

areas are then displayed (via projection) on the work piece surface where users can edit them, C4) users both mark the defects and edit the detected areas on the monitor. All the participants signed a consent form that informed them about the use and confidentiality of their data. They also received a reimbursement for their time.

### 3.1 Task description

First, users mark the damaged spots with a physical tracked pen on either the ceramic surface (C1 and C2) or on the monitor which shows a real-time recording of the basin (C3 and C4). These annotations can be removed at any point by using the user interface to switch the pen from "draw mode" to "delete mode". After finishing the marking process, users can press a button to instruct the robot to scan the basin for annotations. This button is either projected onto the virtual table next to the work piece or displayed on the virtual monitor, depending on the condition.

The robot uses a camera system to detect the markings and visualises the detected areas on the surface of the work piece (C1 and C3) or the monitor (C2 and C4), in order to communicate its interpretation of the user inputs back to the user. These detected areas are displayed as blue rectangles and indicate areas of the basin that will be polished by the robot. Users can remove these blue areas in case of a false detection. As an example, Figure 1 shows two detected areas that are directly projected onto the surface of the basin in C1. Users are able to select and edit various parameters (e.g. pressure, trajectories) for each individual spot. Figure 2 shows the user interface for adapting the parameter settings on the monitor, as utilized in C2 and C4.

After adapting the parameters, users confirm the detected areas and the robot arm polishes all detected spots. Users are able to repeat this process of annotating spots, setting parameters and polishing areas until they cannot find additional spots to repair. At this stage, users finish the task and a new basin with new, randomized defect spots will appear.

Each trial condition had a time limit of 10 minutes. In this timeframe, the goal of the participants was to complete as many rounds as possible, by polishing and delivering every work piece with a flawless surface, acting quickly but accurately to move to the next washbasin as soon as possible.

### 3.2 Technical Implementation

The VR application is implemented in the game engine Unity (version 2022.1.15f1) including the High-Definition Rendering Pipeline (HDRP) to provide high-fidelity graphics. As a model for a polishing robot that would be able to reach all corners of the large (1,8 x 2,4m) ceramic basins, we used a 3D-representation of the Universal Robots UR10 (https://www.universal-robots.com/de/produkte/ur10-roboter/, see Figure 3).

Since the task of the study required the visual detection of scratches, a VR headset with a high resolution was necessary. The Varjo XR-3 in combination with the required software Varjo Base and Varjo Lab Tools was chosen as the target head-mounted display (HMD) for the study. Additionally, the Varjo XR plugin was used for developing for the HMD, and the OpenXR plugin was added to enable tracking of HTC Vive trackers with SteamVR. The physical pen that users used to annotate and interact with the system was

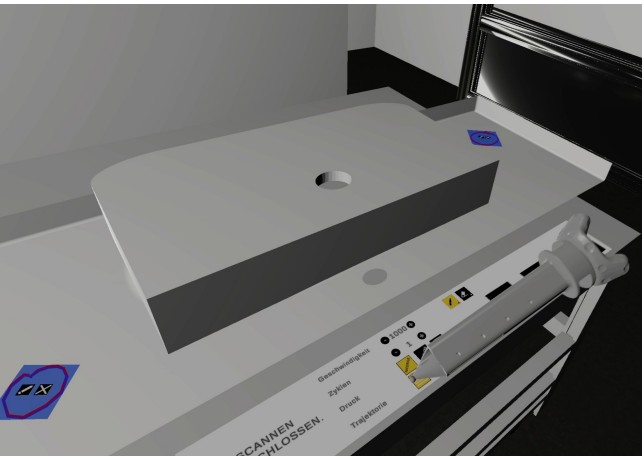

**Figure 1: After marking defects (red circle), blue rectangular areas appear, representing detected polishing areas for the robot.**

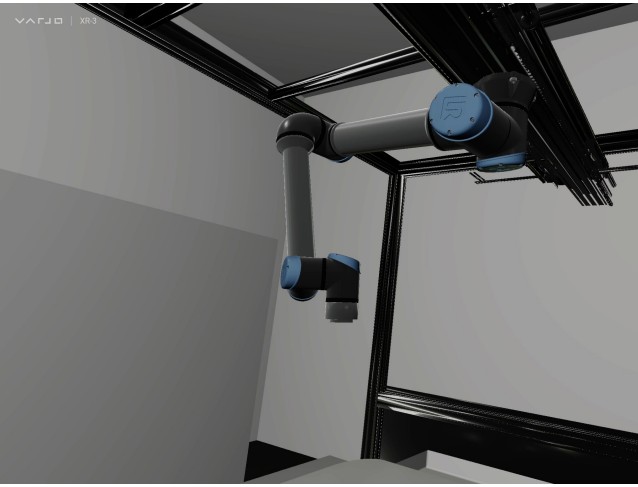

**Figure 3: The robot arm, based on Universal Robots UR10, in charge of polishing based on user instructions.**

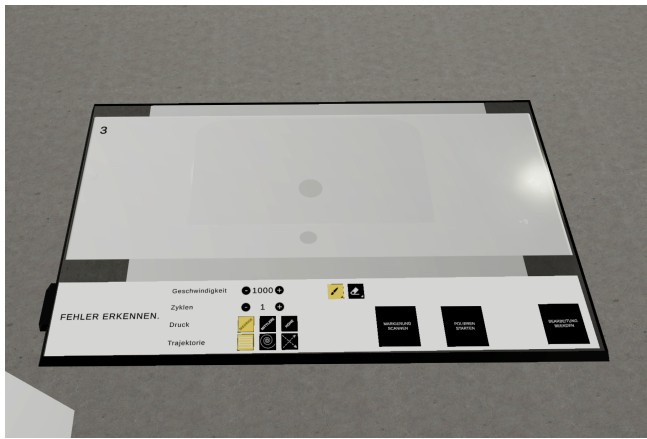

**Figure 2: User interface for editing the polishing parameters of the robot, where the polishing speed, the number of polishing cycles, the pressure and the type of trajectory can be adjusted for each area.**

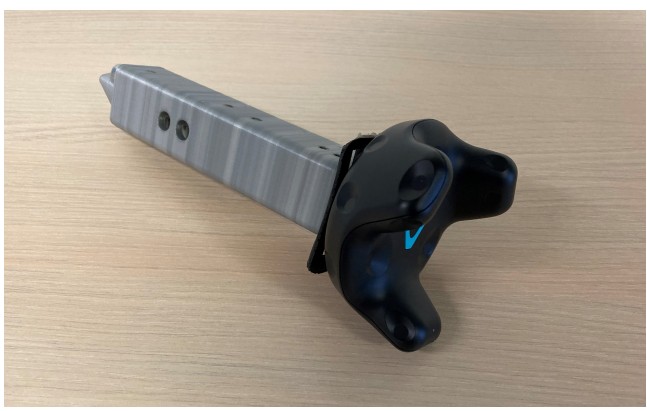

**Figure 4: 3D-printed physical pen with attached HTC Vive tracker**

3D-printed and attached to a HTC Vive tracker for tracking the position and rotation in the virtual environment (see Figure 4).

### 3.3 Participants

Overall, 28 users participated in our VR study, separated into two groups. We conducted a pilot study with the first group. Group 1 (female=1, male=4) included skilled workers in the context of grinding and polishing who conducted the experiment in their factory building in Upper Austria. The mean age was 40 years (SD=13) and none of them had previous experiences with VR. The main study was conducted by the second group. Group 2 (female=7, male=16) included users with technical background e.g. graduated from technical study (but no professional background in the context of grinding and polishing), who conducted the experiment in our

Extended Reality Lab in Vienna. The mean age of this group was 38 years (SD=13) and 19 out of 23 participants already had previous experiences with VR applications. The study setup was identical for both groups.

Regarding previous experiences with robotic systems, Table 1 shows the results of group 2: It can be seen that there is some usage background in the application fields regarding cleaning robots, inspection & quality control robots, packing & palletizing and handling & picking robots. In contrast to this, in group 1 only one person used once a cleaning robot and beside that no familiarity with industrial robots was reported.

### 4 FINDINGS

The overall reaction about performing this task in a virtual setting was positive, i.e., participants quickly get used to it and described the setting as easy (P08, P09, P11). Users indicated that they had no difficulties in understanding the current state of the system and they were able to clearly figure out which task should be carried

| | never used [%] | used once [%] | several times [%] |
|---|---|---|---|
| Assembly & dispensing robots | 91 | 9 | 0 |
| Handling & picking robots | 78 | 13 | 9 |
| Machining & cutting robots | 92 | 4 | 4 |
| Soldering robots | 96 | 4 | 0 |
| Casting/molding robots | 96 | 4 | 0 |
| Finishing robots | 91 | 9 | 0 |
| Logistics & storing robots | 91 | 5 | 4 |
| Packing & palletizing | 74 | 17 | 9 |
| Cleaning robots | 61 | 26 | 13 |
| Inspection & quality control robots | 74 | 22 | 4 |
| Painting and coating robots | 100 | 0 | 0 |
| Harvesting robots | 100 | 0 | 0 |

**Table 1: Familiarity with industrial robots separated by application fields (Group 2).**

out. Furthermore, the graphics were detailed enough and users were not impeded by latency as they were able to detect defects and complete the tasks as required. However, via observations and interviews, we were able to detect some drawbacks in our setup that should be addressed in future studies.

### 4.1 Tracking accuracy

The pilot study was conducted on the premises of a company specialized in the creation of ceramic basins, and involved skilled workers familiar with the challenges of the task at hand. This allowed us to evaluate our suggested solutions in advance with potential future users of the system. However, the conduction of the test in this new environment also resulted in unexpected problems. The first user experienced multiple issues with regards to the tracking accuracy of the pen, which occurred due to the high amount of reflective surfaces (mirrors, reflectors on lamps, polished ceramics etc.) in the area where the study was set up. While these problems were identified and subsequently fixed, it resulted in major time loss and the necessity of ad-hoc adjustments in the room where the study was being conducted.

Furthermore, as the area where the study was being conducted was visible for people walking through the hallway, workers would occasionally stop to observe the study. Discussions between the workers could sometimes be heard from the study area itself, and required the study director to intervene in one case to avoid distractions during the task performance.

As the main study was conducted in a controlled lab environment, these problems no longer occurred in group 2.

### 4.2 Resolution and ergonomics

During the execution of the experiment in the laboratory setting, the test assistant took notes based on the observations and the verbal comments of the participants.

Several participants mentioned, that the virtual monitor is too small (P01, P02, P15, P18 and P24). In particular, P01, P02, P12, P17, P19, P24 and P25 mentioned that detecting smaller defects was difficult due to the small monitor size and P15 stated that reading labels on the monitor was difficult. Furthermore, 11 out of 28 participants explicitly mentioned the poor resolution as a reason to dislike the monitor conditions.

Some interviewees mentioned, that the virtual monitor should be placed at a higher level (P01, P02, P12, P15, P20 and P24). Other suggestions with regards to interface are including the possibility to zoom-in/out on the virtual monitor (P01, P09) and to hide the editing buttons when they are at inspection stage (P07, P34).

In conditions 2 and 3, in which a switch between work piece and monitor interactions is required, it was stated by several participants that the change between the monitor and the work piece was a problem. P01 called the process "annoying and exhausting" (P01). P15 stated that moving from monitor to workbench was physically demanding, stressful for the eyes and had a negative impact on orientation.

In general, most of the participants were fine with wearing the VR headset. However, there were various complaints regarding the weight of the VR headset: P03 mentioned during her final condition to complete, that the headset is heavy. Participants P05 and P08 decided to work while seated during 3 of 4 conditions due to the weight of the headset (the tasks required users to look down most of the time when they were standing). P17 also reported problems while looking down, likely also due to the weight of the headset. P20 stated that sitting felt better. P24 was relieved to remove the headset at the end of the study due to the weight of the headset.

During the interviews, almost half of those who were interviewed indicated explicitly that wearing the VR Headset for completing four rounds of experiments was exhausting. Wearable interfaces are thus still negatively affected by hardware limitations that reduce their usability.

### 4.3 Aesthetics and workload

Upon completion of the last condition, the study participants answered several questions in an interview situation with the test assistant .

A small number of respondents commented about aesthetic parameters and color of virtual environment. For instance, P12 said: "In general I found the room somehow very white and that was a bit exhausting in VR ... everything looks very bluish inside...That's why it doesn't really look that nice. If there were more real colors in the room, it would be more pleasing to the eye".

While performing the required task in VR setting seemed physically easier than real world, the mental workload was not particularly prominent in the interview data. As P01 commented: "...it [collaborating with robot for polishing washbasin] was less physically demanding with VR and safer. However, mentally, and regarding the fact that one has to think, I think it's exactly the same."

## 5 RECOMMENDATIONS

Overall, conducting VR-user studies in the context of HRI is a promising way to analyze various interaction scenarios with lower costs compared to the installation of a real-life robot testing station. Ideally, this allows finding the optimal setting regarding human-robot interaction in an efficient way, and to iterate on promising approaches. However, based on our findings we want to derive some first general recommendations for researchers who plan to conduct

a VR-HRI user study. In general, all recommendations should be considered based on the contextual requirements.

## 5.1 Make the robot's intentions transparent

For VR-based HRI studies, we recommend to put a strong emphasis on clear communication of the robot's intention as well as its perception of the human inputs. This improves both the trust of the workers in the robot and allows for adjustments in case of misinterpretations of the human inputs [10]. This is especially relevant in VR, as the study duration is usually quite limited (due to concerns regarding ergonomics and cyber-sickness), and since users often only have more limited ways of interacting with the robotic system (based on the implemented interactions in the prototype system).

In our VR study, most participants reported that the current state of the system and their next task were clear and understandable. We used an already familiar and very deliberate input method (circling areas that should be polished) to communicate with the polishing robot. To ensure that the human worker is in control, the robot made its interpretation of the polishing areas visible by projecting blue rectangles onto the circled areas, which could easily be adjusted or removed (in case of wrong detection) by the worker. We recommend utilizing similar methods to visualise the next action(s) that the robot will be performing where applicable, in order to reduce the risk of 'misunderstandings' between the human worker and robotic agent.

## 5.2 Provide a well-controlled test environment

VR Studies that include object tracking require a minimum degree of precision, and thus suffer especially from tracking inaccuracies, like the ones we experienced due to the reflections in the test environment during the pilot study. While studies in the laboratory provide sufficient control for researchers and thus make it easier to avoid these issues, studies in the field are crucial to gain valuable insights, but allow for less control of the test environment in advance. Thus, especially for studies with regards to human-robot interaction where high levels of precision are required (e.g. programming-by-demonstration approaches), it is paramount to plan for in-depth pre-testing of the VR setup in order to identify and solve potentially disturbing factors (regarding reflections, noise, etc.), as even minor details can make a big difference in the experience of the user.

## 5.3 Keep ergonomics and exhaustion in mind

While cyber-sickness was less of a problem during our study, multiple users reported issues with regards to the weight of the headset itself. This seems to be consistent with other VR-based user studies (e.g [14, 19]) that wearing the VR headset might cause discomfort and irritations. Thus, physical stress and exhaustion due to the weight of the headset need to be considered. We recommend to use lighter headsets where possible, and/or to reduce the overall time of wearing a headset. Since many of the tasks that users performed during our study were done while looking downwards, the headset likely put additional strain on the neck muscles of the users, which might have led to higher fatigue. Thus, especially for VR studies that involve users working while looking up/down a lot, either *shorter test duration times and/or more breaks* should be planned to give users enough time to rest between tasks.

In our study, we skipped some of the time-consuming, repeating animations of the robot (e.g. the polishing animation), opting for a short fade-out between rounds instead. This allowed us to condense the testing time to 10 minutes while allowing time for differences in task performance among the conditions to emerge more easily. However, this potentially comes at the cost of reduced immersion with regards to the human-robot interaction itself, and should thus be evaluated on a case-by-case basis depending on the goals of the specific study.

In addition to the weight of the VR-headset, switching between two areas in the VR setting can make interactions stressful, annoying and exhausting. We recommend to limit these kinds of switches to a minimum or design the study in a way that switching is not needed, especially for studies that require longer amounts of time spent in VR.

Finally, we also recommend to allow the study participants to adjust the height and the orientation of interactive elements (e.g. the workplace, desk height, height of the display for interaction with the robot etc.) to increase ergonomics. As the physical attributes of users can vary greatly, providing means to adjust these to the users themselves can help alleviate ergonomic issues, improve realism, and ensure that the users can focus on the task at hand.

## 5.4 Make elements easy to distinguish and realistic to look at

Depending on the VR headset used, attention should be paid to the readability of both the user interface and other relevant elements of the virtual environment. The nature of the task at hand during our study (finding scratches on a displayed surface on a virtual monitor in VR) resulted in issues regarding visibility in the monitor-based conditions, despite us using a high-end VR headset (Varjo XR-3) and a regular screen size (21") for the virtual monitor. Similarly, if users are meant to interact on a smaller monitor (e.g. one that is typically attached to a robot arm), the user interface displayed on it should ensure a big enough font size to support readability based on the resolution of the selected VR-Headset.

With regards to immersion, we also recommend to put effort in providing a realistic appearance of the surrounding area. While the 3D-model of the robot was perceived as quite realistic, in our case the colors of the surrounding room were occasionally seen as unrealistic (due to their clean white color) and caused minor irritations for a few participants. On the positive side, scene lightning can support the immersion: We recommend the use of High-Dynamic Range Images (HDRI) for lighting and appropriate reflections to accurately simulate the specific context (in our case, this positively impacted visibility of defects on both the work piece surface and the monitor).

## 6 CONCLUSION AND OUTLOOK

In this paper, we presented recommendations which should support researchers who plan to execute VR user studies in the context of HRI. Based on our VR user study about robotic polishing processes, we explained why controlling the test environment, keeping an eye on ergonomics and exhaustion, as well as clear communication of the robot's actions and understanding is key to derive reliable results in VR user studies. Our study participants were skilled workers

and users with technical background and varying levels of experience with robots. As a next step, we plan to analyze the specific findings our VR study, e.g., which condition outperformed the other ones regarding task performance, accuracy, user experience and task completion time.

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
