# OpenReview forum: "Unlocking the Potential of Virtual Reality in Human Robot Interaction: Insights for User Studies"
_humanrobotinteraction.org/HRI/2023/Workshop/VAM-HRI — VAM-HRI 2023 Oral_

### Official Review · Program_Chairs · 2023-02-26
**Accept**

**Rating:** 6
**Confidence:** 5

**Review:**

Review 1:

This paper presents a set of takeaways for conducting HRI studies in VR. It begins by describing a VR study involving an interaction between a human and robot performing a collaborative ceramic inspection and polishing task, with experimental conditions varying whether different actions were carried out on the surface of the workpiece itself or using a virtualized monitor interface. The experiment was divided into two groups - one performed with expert participants in the field of industrial grinding and polishing in a factory building, and the other performed with novices in a controlled laboratory environment. Feedback from both groups were synthesized, and used to generate a set of general-purpose recommendations for the design of VR-centric HRI experiments.

Strengths:
- The presentation quality and writing style are very clear and easy to read.
- The VR study seems well motivated and well designed. Conducting a specific pre-study with experts in the field seems like it would provide valuable insight. Though no results are reported in this paper, the authors mention this specifically as future work.
- The overall design takeaways are potentially useful for anyone in the HRI community interested in running experiments in VR.

Weaknesses:
- It is odd that no direct results are reported from the experiment described in Section 3 in the paper. While the findings and subsequent recommendations could be considered contributions, they seem mismatched with the experiment, almost as if two papers were spliced together at the midpoint. The first experiment does not seem to be purpose-designed to collect the type of feedback that was reported on in this paper, but rather to compare different modalities of human interaction with a robot in VR. Given that this is the case, it would be more appropriate to report on the results of the actual experiment, condition by condition, before providing the focal analysis regarding generalized feedback and takeaways for VR experiments, even if it is done briefly and if the results do not yield significance yet.
- I think an additional figure or two might be helpful to showcase the different experimental conditions, or to better illustrate some of the points made in the Findings and Recommendations section by showcasing a relevant example from the VR environment.

Despite the lack of direct results from the primary study, it is clear that this work is preliminary, and this paper contains insights potentially valuable to the VAM-HRI research community. Therefore, I recommend acceptance and inclusion in the program.

--------

Review 2:

The motivation for the paper is that testing interactions with robots (particularly large, industrial robots) can be difficult and expensive. The authors propose using VR as a means of facilitating HRI without the need for a physical robot. The authors tested a VR HRI setup using both experts and novices, with four conditions of where the user views and marks defects.

I agree with reviewer one on the lack of quantitative results in the submission, since most of the takeaways are from interviews. There is also not much discussion about differences between the four conditions or differences between novices and experts. To further improve the paper, the authors could more clearly tie together their introduction with a discussion of how the included study intends to illuminate or solve common design problems in VR HRI experiments.

---

### Decision · Program_Chairs · 2023-03-02

Accept (Oral)